# Understanding the Light-Driven Enhancement of CO_2_ Hydrogenation over Ru/TiO_2_ Catalysts

**DOI:** 10.3390/molecules30122577

**Published:** 2025-06-13

**Authors:** Yibin Bu, Kasper Wenderich, Nathália Tavares Costa, Kees-Jan C. J. Weststrate, Annemarie Huijser, Guido Mul

**Affiliations:** 1Photocatalytic Synthesis Group, Faculty of Science and Technology, MESA+ Institute for Nanotechnology, University of Twente, P.O. Box 217, 7500 AE Enschede, The Netherlands; y.bu@utwente.nl (Y.B.); k.wenderich@utwente.nl (K.W.); n.tavarescosta@utwente.nl (N.T.C.); j.m.huijser@utwente.nl (A.H.); 2SynCat@DIFFER, Syngaschem BV, De Zaale 20, 5612 AJ Eindhoven, The Netherlands; keesjan@innovencio.nl

**Keywords:** photothermal catalysis, Ru/TiO_2_, CO_2_ hydrogenation, DRIFT spectroscopy, CO coverage, heat, charge transfer processes

## Abstract

Ru/TiO_2_ catalysts are well known for their high activity in the hydrogenation of CO_2_ to CH_4_ (the Sabatier reaction). This activity is commonly attributed to strong metal–support interactions (SMSIs), associated with reducible oxide layers partly covering the Ru-metal particles. Moreover, isothermal rates of formation of CH_4_ can be significantly enhanced by the exposure of Ru/TiO_2_ to light of UV/visible wavelengths, even at relatively low intensities. In this study, we confirm the significant enhancement in the rate of formation of methane in the conversion of CO_2_, e.g., at 200 °C from ~1.2 mol g_Ru_^−1^·h^−1^ to ~1.8 mol g_Ru_^−1^·h^−1^ by UV/Vis illumination of a hydrogen-treated Ru/TiO_x_ catalyst. The activation energy does not change upon illumination—the rate enhancement coincides with a temperature increase of approximately 10 °C in steady state (flow) conditions. In-situ DRIFT experiments, performed in batch mode, demonstrate that the Ru–CO absorption frequency is shifted and the intensity reduced by combined UV/Vis illumination in the temperature range of 200–350 °C, which is more significant than can be explained by temperature enhancement alone. Moreover, exposing the catalyst to either UV (predominantly exciting TiO_2_) or visible illumination (exclusively exciting Ru) at small intensities leads to very similar effects on Ru–CO IR intensities, formed in situ by exposure to CO_2_. This further confirms that the temperature increase is likely not the only explanation for the enhancement in the reaction rates. Rather, as corroborated by photophysical studies reported in the literature, we propose that illumination induces changes in the electron density of Ru partly covered by a thin layer of TiO_x_, lowering the CO coverage, and thus enhancing the methane formation rate upon illumination.

## 1. Introduction

Combustion and the utilization of fossil fuels have introduced significant quantities of CO_2_ into the atmosphere [1,2,3,4], leading to increasing global temperatures and causing the acidification of oceans [5,6]. To prevent future emissions of CO_2_, alternative resources and routes to chemicals and fuels need to be developed. Since methane has a higher compatibility with existing energy infrastructure than H_2_, a possible strategy is to use electrochemically produced H_2_, and (air-captured) CO_2_ to sustainably produce CH_4_, following the Sabatier reaction [2,7]:(1)$${CO}_{2}+{4H}_{2}↔CH_{4}+2H_{2}O\;\;\;\;\;\;\;\;\;∆H_{298\;{}^{\circ}C}=-165\;kJ\;{mol}^{-1}\;\;\;\;\;\;\;∆S\;=\;-174\;J\;{mol}^{-1}K^{-1}$$

Methanization of CO_2_ is an exothermic process, and a low temperature favors the thermodynamic equilibrium to CH_4_. However, high temperatures (typically > 250 °C) are required to yield considerable methane production rates [7].

An innovative means for the sustainable conversion of CO_2_ to methane is the employment of photocatalysis [4,8,9,10,11,12]. Semiconductors can be excited by photons, thus creating electron/hole-pairs that could be spatially separated over the distinct components of the material [13]. Provided that they do not recombine, these electrons and holes can subsequently be used for reduction and oxidation reactions, respectively. Titanium dioxide is most often investigated in photocatalysis [14,15], even though UV excitation is required for activation. Doping can be used to introduce new electronic levels, allowing for visible light absorption [16,17,18], but dopants typically induce recombination centers, lowering the photonic efficiency.

A promising emerging technology is the combination of light and temperature in so-called photothermal catalysis, driven by light at mildly elevated temperatures [1,2,4,19,20,21,22], requiring combinations of semiconducting supports and light-sensitive nanoparticles. While plasmonic metal nanoparticles (Ag, Au) are particularly effective [22,23,24,25], interestingly catalytically active particles of Ru have also been reported to be sensitive to photon-activation [26,27,28,29,30,31,32]. For Ru/TiO_2_, UV illumination excites TiO_2_, while, simultaneously, visible photons can be utilized to induce two physical phenomena in the plasmonic nanoparticles: (i) the formation of excited states followed by charge transfer to the conduction band of (n-type) semiconductors, such as TiO_2_ [23], or TiO_x_ overlayers; and (ii) localized heating by the decay of localized surface plasmon resonances (LSPRs) [24,25].

Interesting recent studies employing these principles include the investigation of several Group VIII elements on a light-inactive Al_2_O_3_ support for photothermal CO_2_ reduction to CH_4_ and CO by Meng et al. [26]. From the investigated metals, Rh and, in particular, Ru nanostructures were demonstrated to provide excellent activities for CO_2_ conversion, with selectivity values above 99% towards methane [26]. In the same year, O’Brien et al. demonstrated photothermal CO_2_ methanization over Ru-loaded silicon nanowires [27]. In both the studies of Meng et al. and O’Brien et al., the absorption window of the catalyst is remarkably broad, allowing for light harvesting in the visible region and even in the (near-) infrared. Multiple other reports have been published, demonstrating the suitability of Ru nanoparticles (sometimes proposed to be in an oxidized form) for photothermal methanization of CO_2_ [28,29,30,31,32,33,34]. Photo-methanization by Ru on TiO_2_ was already reported in 1987 (with a more than 60-fold increase in activity when the temperature was raised from room temperature to 90 °C) [35], but the number of studies aimed at understanding of the photo-physics [36], and the mechanistic implications of illumination, is limited [3,5].

In this manuscript, we show that Ru on anatase TiO_2_, prepared by NaBH_4_-mediated deposition, is very effective in the photothermal hydrogenation of CO_2_ by performing transient analysis (light-on, light-off cycles). Furthermore, we provide convincing evidence for changes in the electron density of Ru nanoparticles deposited on anatase TiO_2_ upon illumination, through a thorough analysis and assessment of the temperature- and illumination-dependent position of in-situ-generated CO on Ru/TiO_2_ in DRIFT spectra. We discuss the charge transfer effects upon illumination in relation to analyses of the photophysical properties of Ru/TiO_2_ catalysts.

## 2. Results

### 2.1. Characterization of the Photocatalyst

The actual metal loading of the as-prepared catalysts was determined by XRF, and confirmed, on average, to be the targeted loading of Ru of 1 wt-% on TiO_2_, STO, and SiO_2_, respectively. However, the particle size distribution is different for each support, as evident from the TEM images shown in Figure 1.

Analysis of the particles shows uniform Ru nanoparticles are present on TiO_2_, of which ~80% consists of a size of around 1 nm diameter (Figure 1a), with some larger particles in the 2 nm range (~15%), and finally some smaller particles below 1 nm (~5%) after preparation. Treating the sample at 250 °C in H_2_/CO_2_ under illumination did not significantly change the particle size distribution. Nonetheless, ~70% of the Ru particles consist of a diameter of around 1 nm, although some particles in the 3 nm range are now visible (contributing to less than 5%). Prolonged exposure to 450 °C under illumination in H_2_/CO_2_ did not lead to significant changes either, with ~65% of the Ru particles consisting of a diameter of around 1 nm. Ru/TiO_2_ catalysts have been previously synthesized and analyzed in a multitude of studies aimed at catalytic hydrogenation of CO, CO_2_, or mixtures thereof. Several recent studies aimed at achieving a better understanding of the complex interactions between Ru nanoparticles (NPs) and oxide supports (typically TiO_2_), typically assigned to strong metal–support interactions (SMSIs) associated with structural modifications such as layers of TiO_x_ on Ru particles [29,30]. In Appendix A, we show images of the Ru/TiO_x_ sample after H_2_ treatment, which are similar to the particles reported by Zhang et al. [37] and Abdel-Mageed et al. [38]. The overlayer is indicated by the blue arrows inserted in the picture (Appendix A). Finally, Abdel-Mageed and coworkers have been able to determine the size distribution of Ru nanoparticles on P25 TiO_2_ at a high resolution [38], which unfortunately was not feasible using the acquired images in this study.

After H_2_ treatment at 450 °C, inhomogeneous and bigger Ru particles are observed on STO and SiO_2_, respectively. On STO, only approximately 35% of the particles are present in the size range of 1 nm, while 50% of the particles contain diameters of ~2 nm (Figure 1d). On SiO_2_, the Ru particle size distribution is even less homogeneous, with sizes extending from 2 to 6 nm (Figure 1e). In both cases, evidence for overlayers of oxides is absent, which is likely associated with limitations in reducibility (for SrTiO_3_), or the absence of reducibility of the oxides (for SiO_2_).

X-ray photoelectron spectroscopy studies were performed to study the surface oxidation state of the Ru/TiO_2_, Ru/SrTiO_3_ (STO), and Ru/SiO_2_ catalysts after preparation, including H_2_ treatment at 450 °C; the spectra are shown in Figure 2a. For all three catalysts, the intensity profile can be deconvoluted by three peaks. The peak at ~284.8 eV (green trace) is attributed to adventitious carbon species on the surface. The peaks at 283.8 eV and 279.6 eV are assigned to metallic Ru (Ru 3d_3/2_, yellow trace, and Ru 3d_5/2_, blue trace, respectively), demonstrating that the preparation procedure—using NaBH_4_ and 50 °C for 2 h—results in the complete reduction of the Ru precursor salt. The spectra are comparable to spectra reported by Abdel-Mageed et al. [38] and Cisneros et al. [39], but seem to contain significantly less carbon contaminant, suggesting that the washing and reductive treatment is very affective in the removal of carbon-containing residues. Figure 2b shows the UV-Vis absorbance spectra of the catalysts. The pure TiO_2_ (anatase phase) nanoparticles only exhibit strong absorption below 400 nm, corresponding to a bandgap of 3.0 eV (slightly lower than the expected 3.2 eV bandgap) [15], while the Ru-containing catalysts additionally show a broad absorption band from 400 to 1000 nm, with a shallow optimum at around 500 nm, which is particularly visible for the Ru/TiO_2_ sample. Similar behavior can be observed for SrTiO_3_ (bandgap of 3.1 eV in this study, in line with the expected 3.2 eV) [15]. The nature of the strong visible light absorption of Ru is still under debate. In some studies, the intensive broad absorption from 400 to beyond 800 nm is assigned to RuO_2_ absorptions [29,30]. Other research groups suggest that light attenuation by small, well-dispersed Ru nanoparticles is responsible for the strong light absorption [27,28]. Given that the XPS spectra conclusively show the presence of Ru in the metallic state, and the observed size of the Ru^0^ particles in Figure 1 is in the order of 1 nm, we favor the assignment of strong light absorption at visible wavelengths to well-dispersed Ru^0^ nanoparticles. The width of the absorption band centered around 490 nm likely results from structural inhomogeneity, as the Ru nanoparticles show a distribution in sizes (see Figure 1).

### 2.2. Photothermal Activity—Transient MS Analysis

We will now discuss the performance of the three supported and reductively treated Ru catalysts in a relatively low temperature range, where CH_4_ formation is not significantly thermodynamically limited.

Figure 3 and Figure 4 show the CH_4_ formation rates for a stepwise increase between 120 °C and 220 °C (see top of the curve). The bare TiO_2_ support is inactive for the production of methane in the investigated temperature range, both in the dark and under illumination. Ru nanoparticles are clearly necessary to induce the formation of CH_4_ by conversion of CO_2_ and H_2_. The activity of Ru/TiO_2_ is significantly larger than that of Ru/STO and Ru/SiO_2_, which is in agreement with the literature investigating catalytic performance of (TiO_2_)-supported Ru catalysts, typically assigned to strong or electronic metal–support interactions. For example, at 220 °C, the Ru-normalized CH_4_ formation rate of Ru/TiO_2_ amounts to 1.8 mol g_Ru_^−1^ h^−1^ under UV-vis irradiation, which is 4.5 times higher than that obtained for Ru/SrTiO_3_ (STO, 0.4 mol g_Ru_^−1^ h^−1^). In agreement with the obtained (low) rate for Ru/STO, Mateo et al. report for STO-supported—oxidized—RuO_2_ nanoparticles (which likely are reduced in situ in the process conditions) a steady state production of 50 mmol g_Ru_^−1^ h^−1^ at 150 °C under UV-vis irradiation (1 sun, 300 W Xe lamp) [29]. As shown in Table 1, Wang et. al. report an activity for Ru/TiO_2_ in the formation of CH_4_ of 172 mmol g_Ru_^−1^ h^−1^ at 150 °C under 1 sun irradiation (AM 1.5). They also report stability of the system for at least 1000 min at 200 °C in the photothermal reduction of CO_2_ [5]. Finally, Novoa-Cid et al. demonstrate that small Ru nanoparticles (1.5 nm) supported on Titanate nanotubes produce CH_4_ at a rate of 110.7 mmol g_Ru_^−1^ h^−1^ at 210 °C in a pressured batch reactor under UV-vis-NIR illumination (1 sun) [30]. The strongest light enhancement is observed around 200 °C in Figure 3, where the rate rapidly increases from ~1.2 mol g_Ru_^−1^ h^−1^ to ~1.8 mol g_Ru_^−1^ h^−1^ upon illumination.

In the following section, we will address the performance of Ru/TiO_2_ at higher temperatures, in the temperature range of 200 °C to 450 °C, where thermodynamic limitations in the formation of CH_4_ begin to play a role. Figure 5 shows the transients in CO_2_ (blue), CH_4_ (red), CO (orange), and H_2_ (black) of the ion currents recorded by the Mass Spectrometer, as a function of temperature and a transient in light-on, light off cycles. Upon light-on, up to 300 °C, a positive response of the CH_4_ signal to light is observed. The response of the formation of CH_4_ to light is most dominant at 250 °C, as further illustrated in Appendix A—comparing the response of the formation of CH_4_ to light at 200, 250, and 300 °C, respectively. It should be noted that, at 300 °C, conversion of CO_2_ and H_2_ to CH_4_ becomes significantly affected by thermodynamic equilibrium [30].

Thus, at higher temperatures than 250 °C, the light effect on CH_4_ production diminishes, interestingly to become negative at 450 °C. This is obviously because the CO_2_ conversion to CH_4_ is now significantly limited by thermodynamic equilibrium [38]. At this point, exposure to light shows a promotion in the formation of CO—an intermediate in the formation of CH_4_—with a concomitant decrease in the production of CH_4_, again in agreement with a (local) temperature increase. At each temperature, the light-promoted formation of CH_4_ or CO can be well reproduced. Lowering the temperature shows the high stability of the catalyst, since the light-induced transients and the temperature-determined quantities in CH_4_ are comparable to the amounts obtained in the increasing temperature curve. In other words, a significant hysteresis in performance of the Ru/TiO_2_ catalyst at all temperatures and reaction conditions is absent under the process conditions investigated here. In Appendix A, enhancement in the production of CH_4_ is compared to the Ru/TiO_2_ catalyst exposed to UV/Vis or Vis light (using a cut-off filter for wavelengths < 420 nm), respectively (See Appendix A for light emission spectrum). UV wavelengths apparently do contribute significantly to the promotion of the conversion in the applied flow reactor. This implies that light absorption by the Ru particles is predominantly responsible for the promotion in the rate of formation of CH_4_.

The high photothermal stability of Ru/TiO_2_ was further investigated at 250 °C under chopped UV-vis irradiation for a total period of 24 h (Appendix A). The CO_2_ conversion amounts to 30% at 250 °C in the dark, which increases to 40% under illumination. A corresponding (higher) increase in H_2_ conversion can be observed. Appendix A also demonstrates the excellent stability of (H_2_-treated) Ru/TiO_2_ in time at 250 °C for at least 24 h. Notably, in the work of Garcia and coworkers, the performance of Ru(O_2_)/STO decreases significantly under UV-vis illumination in a period of 3 h [29]. Hence, the H_2_-pretreated Ru/TiO_2_ catalyst in our study not only shows a high mass-based activity, but also a high stability in performance, probably related to the strong metal–support interaction achieved by the H_2_ treatment. We assign the increase in conversion of CO_2_ (and H_2_) predominantly to an increase in the catalyst temperature of 10 °C upon illumination. To corroborate this increase in temperature, we also provide some back-on-the-envelope calculations on page 9 of the SI, confirming that such an increase is entirely possible considering the light intensity (360 mW/cm^−2^) exposed to the sample. We hypothesize that the increase in CO_2_ conversion and CH_4_ productivity is due to a reduction in Ru–CO surface coverage. To analyze the effect of illumination on the Ru–CO coverage, we will now discuss the results of in-situ DRIFT measurements, as well as extensively examining the assignment of the observed IR absorption bands.

### 2.3. DRIFT Analysis

The results of the DRIFT (static gas—no flow) measurements are shown in Figure 6 (for Ru/TiO_2_), using the same light source as used for the flow experiments illustrated in Figure 3. It should be noted that the illumination measurements need to be considered in situ (and not operando), since the spectra were recorded in the absence of flow (the gas composition equilibrates to a certain composition—please consider that H_2_ is the limiting reactant (a ratio of CO_2_:H_2_ of 1:3 was applied—see reaction (1)). After H_2_ pretreatment, CO_2_ gas was introduced into the cell at 50 °C, and infrared bands at 1628, 1566, and 1363 cm^−1^ become apparent—of which the intensity stabilizes after 30 min. These bands are very similar to the bands obtained in the experiment with the bare TiO_2_ support (see Appendix A) and have been assigned to carbonate and carboxylate species of different conformation (monodentate, bidentate) [40,41], adsorbed on titania. The small band at ~1420 cm^−1^ can be assigned to carbonate. In addition, for Ru/TiO_2_ (Figure 6), bands of low intensity appear at 2072 and 2018 cm^−1^, which can be attributed to surface-adsorbed CO on partly-oxidized Ru and metallic Ru, respectively, likely coordinating via the carbon atom [41]. The observation of CO adsorption suggests that the Ru particles are incompletely covered by TiOx, and significant interaction of Ru^0^ with the gas phase is possible. This adsorption also seems to indicate that dissociation of CO_2_ into CO takes place at 50 °C, likely accompanied by partial oxidation of the Ru particles, explaining the presence of the 2072 cm^−1^ band [40]. After introduction of H_2_/CO_2_, and stabilizing for 30 min, the peak at 2072 cm^−1^ almost completely disappears (in agreement with oxide reduction) and the intensity of the band at 2018 cm^−1^ significantly broadens and increases. Obviously, hydrogen promotes the conversion of CO_2_ to (surface-adsorbed) CO, without significantly changing the intensity of the carboxylate peaks—only the peak at ~1420 cm^−1^ seems to decrease by the introduction of hydrogen. The effect of UV-Vis irradiation on the reaction is quite limited at 50 °C, showing few spectral changes.

At 250 °C, similar peaks to those observed at 50 °C can be observed after CO_2_ treatment, with a slightly higher intensity of the Ru^0^-CO band at 2018 cm^−1^. Introducing H_2_ into the CO_2_ gas flow again causes the 2018 cm^−1^ band to significantly broaden and increase in intensity. The rotational spectral signature of a band at 3017 cm^−1^ is attributed to CH_4_ gas [42]. UV-vis irradiation greatly enhances the intensity of the CH_4_ signature. Clearly, UV-vis irradiation promotes the conversion of CO_2_ (and the intermediate CO) into CH_4_ by reaction with H_2_, which agrees well with the activity tests. Most importantly, the Ru–CO band seems to decrease in intensity upon illumination, and significantly shifts to lower wavenumbers as a result of decreasing coverage with CO.

At 350 °C, a much higher quantity of CO on Ru appears at 2026 cm^−1^ (Ru^0^-CO) already in a CO_2_ environment in the absence of H_2_, in comparison to the low intensity CO signature at 50 and 250 °C. This suggests that 350 °C is sufficient for significant CO_2_ dissociation. Similar to 250 °C, introducing H_2_ into CO_2_ results in CH_4_ gas formation, but does not dramatically change the peak profile of adsorbed Ru^0^-CO. Upon irradiation with UV-Vis light, the rotational CH_4_ signature now decreases slightly in intensity and gas-phase CO can be observed (in the range of ~2100 to 2200 cm^−1^) [43]. Again, a significant decrease in the Ru–CO intensity can be observed upon illumination, accompanied by a shift to a lower wavenumber. At 450 °C, the observations are similar to those at 350 °C, with illumination lowering and shifting the CO band intensity. The amount of gas-phase product is not significantly affected, in agreement with the transient MS analysis.

In addition to illumination with UV/Vis radiation, we also exposed the catalyst in the IR Cell either to exclusively UV (370 nm) irradiation or to visible irradiation (550 nm)—see Appendix A. The development of the spectra of the catalyst when exposed to CO_2_ in the dark is very similar to that observed in Figure 6, with the CO band on both oxidized and reduced Ru clusters obvious at 50 °C, as well as the formation of (bi)carbonate. At an elevated temperature (250, 350, and 450 °C, respectively), the broad band of CO adsorbed on Ru^0^ clusters develops. The introduction of H_2_ at these temperatures again results in the formation of methane, accompanied by a decrease in intensity of the band of adsorbed CO. Surprisingly, in the presence of H_2_, illumination with 370 (Appendix A) or 550 nm (Appendix A) does not now significantly affect the position and intensity of the IR band assigned to CO adsorbed on Ru^0^ nanoclusters, nor the intensity of the methane signature (band at 3017 cm^−1^). Interestingly, differences in the absence or presence of illumination can be observed in the absence of H_2_, where the effect of UV radiation and visible light excitation (550 nm) is most pronounced at 450 °C (Appendix A).

## 3. Discussion

### 3.1. Metal–Support Interactions

As stated in the introduction, the high activity of Ru/TiO_2_ in methanation of CO_2_ is explained by the so-called SMSI effect, which has been extensively discussed in the literature [38,39,40]. Close inspection of the comparison between different supports shown in Figure 3 reveals that titania, known for this effect, results in dramatically higher performance than the other supports—and TEM characterization confirms that the preparation procedure, followed by treatment in H_2_ atmosphere, leads to the formation of an amorphous overlayer, only for the Ru/TiO_2_ catalyst—which is predominantly responsible for the higher activity (see SI Appendix A). The temperature of 450 °C in H_2_ atmosphere used here is high enough to induce titania migration onto the ruthenium particles, as also shown by Xu et al. [44]. We acknowledge that the particle size determination in TEM, combined with the assumption of spherical (or hemi-spherical) particles, yields a three-times higher dispersion on titania compared to silica, while the dispersion on STO is 1.7 times higher than on silica, but we assume this has only minor consequences for the observed rate [38]. The performance variation between Ru/STO and Ru/SiO_2_ could stem from support oxygen vacancies in STO, which would require further XPS valence state analysis and oxygen defect analysis of the STO (SrTiO_3_) used [29]. Ru/SiO_2_ is known to show little (photo)catalytic activity, due to the lack of SMSI effects [28,29,30,31,32,33,34,35,36,37,38,39,40], and the absence of charge transfer phenomena between Ru particles and the SiO_2_ support upon illumination of Ru [36].

In our IR studies, we observed a significant quantity of CO adsorbed on ruthenium metal, showing that TiO_x_ likely does not fully cover the surface. We therefore envision that Ru^0^ clusters are in strong interaction with (reduced) TiO_x_ sites, (with some of) the reaction steps likely occurring at sites located near the overlayer or present at the perimeter between TiO_x_ and Ru.

### 3.2. Surface Coverage—Assessed by Literature Evaluation

For the description of the reaction rate and reaction selectivity, we need to consider the surface-mediated reactions given below, where * denotes a vacant site on the Ru surface and (H) indicates the reactions in the presence of H_2_:CO_2_ (g) + * ↔ CO_2 ad_(2)CO_2_ (+H) + * ↔ CO_ad_ + O(H)(3)CO_ad_ (+x H) + * ↔ C(H)_ad_ + O(H)_ad_(4)C(H)_ad_ + 3 H_ad_ → CH_4_ (g) + 4 *(5)O(H)_ad_ + H_ad_ ↔ H_2_O (g) + 2 *(6)CO_ad_ ↔ CO (g) + *(7)

Reactions (2) and (3) describe the adsorption and dissociation of CO_2_ on the catalyst surface. CO_2_ adsorption studies on silica-supported ruthenium by Zağli and Falconer [45] show that direct CO_2_ dissociation already occurs at room temperature. In line with this, we do indeed find some adsorbed CO after exposure to pure CO_2_ at 50 °C for Ru/TiO_2_. The amount of CO at 50 °C increases substantially when hydrogen is added, and this may point to a hydrogen-assisted route in which CO_2_ dissociation proceeds. We note that we do not observe formate species in IR, which would appear around 1590 cm^−1^ and which are particularly prominently seen at 400 °C on ~1.8 nm Ni particles on silica [46]. Irrespective of the contribution from hydrogen-assisted routes, we conclude that CO_2_ dissociation is a fast reaction and does not limit the overall reaction rate. This is in line with isotope exchange experiments reported by Mansour and Iglesia [47], who show that the CO_2_ dissociation step is equilibrated on Ru/SiO_2_ at 300 °C and is thus not rate-limiting for the reaction. Our IR measurements at 350 °C under pure CO_2_ show the dynamic equilibrium in action: the spectrum shows a clear band due to adsorbed CO, which we attribute to CO_2_ dissociation (likely concurrently oxidizing oxygen vacancies) at the Ru_TiO_x_ interface.

Likewise, the dissociation of CO, step (4), is not a very difficult reaction. Surface science studies show that step sites [48], as well as other types of undercoordinated sites [49], are active for direct CO dissociation, a reaction that occurs below 127 °C on Ru. More recent work on mass-selected ruthenium nanoparticles [50] as well as an STM study on stepped ruthenium [51] identified undercoordinated sites as the locus where direct CO dissociation can occur. These studies also show that the reverse reaction, carbon + oxygen to form CO, occurs around 277 °C, which implies that Reaction (4) becomes reversible above this temperature. In the previously mentioned study by Zağli and Falconer [45], silica-supported ruthenium was exposed to either CO_2_ or CO at room temperature and subsequently heated in the presence of hydrogen (temperature-programmed hydrogenation, TPH). The onset of methane was found around 127 °C, irrespective of the precursor used, which shows that CO dissociation must have occurred below this temperature already. These authors also deposited carbon and oxygen on the surface by adsorbing either CO or CO_2_ at room temperature and heating to high temperature in an inert atmosphere to induce dissociation. In the TPH performed afterwards, the formation of methane was already found at room temperature, a finding that corroborates the earlier report of methane formation at room temperature by Low and Bell [52] on alumina-supported ruthenium. This shows, firstly, that Reaction (5), hydrogenation of surface carbon, is a very easy reaction that can occur at room temperature already. It also means that the onset temperature of 127 °C for methane formation during the TPH of the CO-covered sample is either determined by the onset temperature of CO dissociation or instead caused by CO poisoning of dissociative hydrogen adsorption.

The same study [52] also informs us about Reaction (6), the removal of oxygen from the catalyst surface. The TPH experiments consistently show that the water formation peak is shifted to 10–15 degrees relative to the methane formation peak, and it also extends to a much higher temperature than the methane peak. Water formation thus requires a higher temperature to proceed than any of the other reactions, which suggests that oxygen removal has the lowest rate constant of all the reactions listed above. This is consistent with the positive order in hydrogen pressure for CO_2_ hydrogenation on ruthenium, as reported by Mansour and Iglesia [47]. In this view, the promoting role of titania is to provide an alternative, slightly easier (5 kJ mol^−1^) route to remove oxygen from the surface.

The adsorbed hydrogen atoms required in Reactions (5) and (6) stem from dissociative adsorption of the H_2_ reactant. The maximum hydrogen coverage is determined by the number of available sites left open for hydrogen to adsorb, multiplied by the fractional occupancy of those sites as determined by the hydrogen adsorption–desorption equilibrium. Considering the hydrogen adsorption energy on ruthenium of around 100 kJ mol^−1^ [53], and the relatively low H_2_ pressure of 300 mbar used here, the hydrogen coverage at the highest temperatures studied, 450 °C, is expected to be significantly lower than at lower temperatures. We therefore attribute the observed drop in conversion above 400 °C and concomitant selectivity change to CO, the less-hydrogenated product, to a decreased hydrogen coverage.

Dissociative hydrogen adsorption (as well as a number of other reactions from the list) requires vacant surface sites that are rather scarce at low temperature since the adsorption of carbon monoxide, the primary product of CO_2_ dissociation, is rather strong. The desorption temperature of around 227 °C on Ru(0001) for CO coverages below 0.33 ML translates to an adsorption energy of 150–160 kJ mol^−1^. CO desorption from ruthenium nanoparticles as well as from more open ruthenium surfaces [49] occurs at a significantly lower temperature. CO desorption from size-selected ruthenium nanoparticles [54] as small as 2.5 nm was found around 127 °C, a temperature that translates to an adsorption energy of 125 kJ mol^−1^ using a Redhead equation with ν = 1 × 10^15^ s^−1^. We will hereafter use this value as characteristic of the CO desorption barrier from ruthenium nanoparticles.

### 3.3. Explaining Temperature-Dependent (Surface) Chemistry

The IR spectrum at 50 °C in CO_2_/H_2_ shows a significant amount of adsorbed CO, which is a clear indication that CO_2_ dissociation is possible at this temperature. However, the adsorption energy of 125 kJ mol^−1^ translates to a CO residence time of (1/k_des_) of 1.5 × 10^5^ s at 50 °C, which means that CO is essentially irreversibly adsorbed and poisons the surface for adsorption of reactants—so that no steady state reaction is possible. Figure 3 shows that the onset of the methanation reaction is around 140 °C, which is essentially the onset temperature where CO starts to desorb from ruthenium nanoparticles.

The CO residence time at this temperature has dropped to 6 s, which is fast enough so that CO adsorption, Reaction (7), has now become reversible. The CO coverage is still very high and blocks the majority of the surface sites, but as there is now a dynamic adsorption–desorption equilibrium, (short-lived) vacancies now randomly appear on the surface so that adsorption and dissociation of hydrogen and CO_2_ is possible, and the steady state reaction can occur.

The large gain in activity seen upon increasing the temperature can be understood as a lowering of the CO equilibrium coverage. This is directly visible in the IR spectra, showing that the intensity of the CO absorption band decreases as the temperature increases. The frequency also shifts to lower wavenumbers with increasing temperature, which can be attributed to a coverage effect. Adsorption studies on silica-supported ruthenium nanoparticles show that the frequency at which adsorbed CO appears depends on the CO coverage, where a higher frequency means a higher CO coverage [55]. Using the combination of intensity and peak position as a probe for the CO surface concentration, we find that the decrease in the CO equilibrium concentration from 250 °C to 350 °C is much larger than a decrease from 350–450 °C. This matches the trend in activity seen in Figure 4, where the step change from 250 °C to 350 °C causes a significant gain in activity while the step from 350 °C to 450 °C actually causes a loss of (methane formation) activity. This shows that CO poisoning is dominant at a lower temperature while it ceases to play a role above 350 °C (under the low-pressure conditions used here). As mentioned before, the surface hydrogen concentration instead appears to become the limiting factor at high temperatures. The conclusion that CO poisoning is rate-limiting for CO_2_ hydrogenation below 350 °C is consistent with the previously mentioned kinetic studies performed at 300 °C on silica-supported ruthenium [47], from which it was also concluded that the rate at this temperature is inhibited by the CO reaction product.

### 3.4. Effect of Photoexcitation on Performance

The experiments show that irradiation has the most prominent positive effect on the Ru particles deposited on TiO_2_. We first of all note that the Arrhenius plot of CH_4_ formation in the dark and under illumination, as derived from the low temperature range reported in Figure 3, yields a similar apparent activation energy, while the pre-exponential factor (Appendix A) is different, indicating that irradiation does not change the reaction mechanism [56]. The absolute gain in CO_2_ conversion is by far the largest at 250 °C, in the temperature window where CO poisoning is the dominant rate-limiting factor. The IR measurements show that irradiation in this temperature window causes a substantial lowering of the CO coverage, and this rationalizes the strong light-induced promotion at this temperature.

The light-induced changes are in fact quite similar to a temperature increase, which raises the question of whether the light selectively deposits heat in the catalyst particles so that their local temperature is about 10 °C higher than the average temperature of the catalyst bed. We can test this hypothesis by using the position and area of the CO absorption band in the IR spectrum as an in-situ probe of the CO equilibrium coverage. The peak position and peak area at 250 °C under illumination approximately corresponds to the position and area found at 450 °C without illumination. In case of a purely thermal effect of light, to explain these spectral data, it would mean that the average temperature of the particles should be 450 °C and one would thus expect to obtain the CH_4_/CO selectivity characteristic of 450 °C under illumination at 250 °C. This is not the case, since at 450 °C significant CO production is observed (see the rotational signature of gas-phase CO), whereas this is completely absent at 250 °C under illumination (see the spectra and product distribution).

Non-thermal weakening of the Ru—CO bond is also achieved at relatively high temperatures in conditions without H_2_, when exclusively UV or green light illumination is applied, while the effect is similar at both wavelengths: see Appendix A. The physical phenomena at play in non-thermal weaking of the Ru–CO could be as follows.

#### 3.4.1. Further Consideration of the Effect of Illumination on CH_4_ Productivity and Ru–CO Surface Coverage

Further considerations for excluding the possibility of the thermal heating effects to account for our spectral data, implying a temperature rise of ~200 degrees, include a back-of-the-envelope calculation of the temperature increase based on complete absorption of all the light energy by the catalyst bed (see Appendix A). We only calculate a maximum rate of temperature increase of ~2.9 K s^−1^, which agrees with the activity data obtained in our flow reactor but does not agree with the 200 degree rise required to explain the differences in CO-infrared absorption intensities and peak positions. Finally, a recent study helpfully illustrates the heating effects of illumination of a catalyst bed [57]. Analyzing the reported temperature increase as a function of intensity of illumination, the increase at the intensity of light used in our study can indeed be expected to be limited to ~10 °C, implying that additional non-thermal weakening of the Ru–CO bond is at play in our infrared experiments.

#### 3.4.2. Non-Thermal Effects

The question remains as to why light leads to such lower CO coverage at intermediate temperatures, if this cannot be entirely explained by heat effects. One explanation can be that the light-induced electric field associated with the Ru particles may polarize adsorbed surface species [58] and reduce CO poisoning [59]. Electron transfer phenomena have also been reported for Ru clusters on semiconductor entities, which are indeed temperature-dependent, in line with the temperature dependence of the lowering of the CO surface coverage upon illumination [60]. In line with this hypothesis, the following situations can be proposed. Although we cannot exclude absorbate–metal electronic transitions as described in the literature, for example for Pt–CO and Ru–CO [61,62], Figure 7a,b illustrates the photoinduced hole (or electron) transfer from a Ru nanocluster towards the TiO_x_, depending on the position of the HOMO and LUMO energy levels of the Ru nanoclusters relative to the valence band and conduction band of the TiO_x_ [36]. The photoexcited electron in the Ru could promote CO desorption (Figure 7b) [63]. A derived third scenario, which is possible due to the hydrogen treatment preceding the photothermal catalysis partly reducing the TiO_x_, is shown in Figure 7c. Photoexcitation of the Ru nanocluster is followed by a hole transfer towards the reduced TiO_x_, while the excited electron promotes CO desorption. Finally, Figure 7d illustrates a fourth possibility, with excitation of the electronic transition between hybridized absorbate–metal bonding and antibonding states destabilizing the metal–CO bond [62]. Considering the major difference in activity we observe between Ru/SiO_2_ and Ru/TiO_2_ and the weak activity of the first (Figure 3), we believe the latter is unlikely to play a significant role here.

Sa and coworkers recently reported phonon-assisted hot carrier generation for Au/ZrO_2_ and Au/TiO_2_ systems, in contrast to the regular thermally deactivated electron transfer observed in non-plasmonic systems [64]. It is important to note here that the Ru nanostructures in this study are very small; consequently, their behavior is likely at the border of plasmonic nanoparticles, and rather agrees with non-plasmonic nanoclusters showing molecular-type properties [65]. Ultrafast (luminescence) studies at in-situ conditions are currently being performed to unravel the temperature-dependency of the light-induced charge separation and recombination dynamics of Ru/TiO_2_ catalysts, and if these processes occur between the Ru nanoparticles and TiO_2_ and/or between Ru nanoparticles and surface-adsorbed species.

#### 3.4.3. Simplified View of Photon-Induced Changes in CO Coverage

A simplified view of the effect of illumination on CO coverage is shown in Figure 8. The left figure shows that UV irradiation results in electron transfer via reduced TiO_x_ at the perimeter, to the Ru nanoparticles, causing enhanced electron density, thereby reducing the surface coverage of CO according to rate constant k_2_—formed by thermal conversion of CO_2_ (according to rate constant k_1_). In the presence of hydrogen, the formation of CO is very fast (k_1_ is large by coproduction of H_2_O); hence an effect on the CO coverage was not observed upon excitation with UV light alone. The right scheme shows that visible irradiation results in hole transfer from Ru to the (reduced) TiO_x_ at the perimeter of the particles, again causing enhanced electron density, and reducing the surface coverage of Ru–CO according to a rate constant k_2_. We would like to state that positive charge transfer from Ru to TiO_2_ is not in agreement with localized surface plasmon resonance (LSPR), which can still create “hot electrons” in the electronic band of the plasmonic metal, which are then injected into the conduction band of the semiconducting TiO_2_. In fact, the transfer of holes in Ru to TiO_2_ is counter-intuitive because of the depletion zone, characteristic of a Schottky barrier at a metal–semiconductor interface, which lowers the concentration of electrons in the TiO_2_ conduction band near the interface. Nevertheless, the observed shift in CO band upon illumination to lower frequencies is in agreement with a larger electron density on the Ru particles. When both UV and Vis illumination are applied, apparently the increase in charge density of the Ru particles is sufficient to even observe an effect on the surface coverage of CO in the presence of H_2_.

While CO poisoning dominates the reaction rate to methane in the range of 200–300 °C, as we have substantiated by citing the relevant literature, several hypotheses can be proposed to explain the limited extent to which the methane formation rate can be enhanced. As we have stated previously (see steps (4) and (5)), other surface adsorbates also play a key role in the mechanism, and non-thermal effects might also cause changes in the Ru–H coverage, for example, which not only affects the CH_4_ formation rate (step 4), but also the oxygen removal rate to form water (step 5). We are currently assessing the options to investigate the effect of illumination on the likely decrease in Ru–H coverage.

We would like to note, finally, that the effect of metal loading on catalyst performance merits further investigation. Increasing the loading could lead to various effects from a (thermal) catalytic perspective: a higher Ru loading leads to more active sites, enhancing performance, but at the same time it could reduce the stability of the catalyst by increasing the probability of sintering. From the perspective of light absorption by the TiO_x_ support, a higher Ru loading could shield the TiO_2_ surface from light, leading to lower UV absorption, and thus possibly a lower effect of illumination on enhancement of the methane formation rate. Studying the effect of loading on performance and light enhancement is part of our ongoing research activities, with a focus on irradiation at relatively low intensities. This is contrary to reports in the literature where light of more than one order of magnitude higher intensity has been utilized, and thermal effects are likely dominant in enhancing rates upon illumination [66,67,68].

## 4. Experimental

### 4.1. Catalyst Preparation

Ruthenium (III) chloride hydrate (0.103 g, Sigma-Aldrich, Burlington, MA, USA, 99.98% trace metals basis) and polyvinylpyrrolidone (0.523 g PVP, average molecular weight 55,000, Sigma-Aldrich) were dissolved in a mixture of methanol (200 mL, Sigma-Aldrich) and MilliQ water (160 mL) in a glass beaker while continuously stirring with a stir-bar. Then, titanium dioxide (1.0 g, Sigma-Aldrich, anatase, 99.7%, <25 nm particle size) was added to the above solution, which was stirred continuously at room temperature for 1 h. Subsequently, sodium borohydride (0.185 g NaBH_4_, Sigma-Aldrich, Purum p.a., ≥96%) was added instantly into the above suspension, which turned dark black immediately. After continuous stirring for 2 h at room temperature, the suspension was heated on a heating plate to 50 °C and stirred for another 2 h. Then, the suspension was filtered with the aid of filter paper. After liquid removal was completed, the filter cake was washed thoroughly with MilliQ water, collected in a crucible, and dried overnight by inserting the crucible in a heating stove kept at 90 °C. The deposition of Ru on SrTiO_3_ (Aldrich) and SiO_2_ (Degussa, Germany) was performed the same way, by the introduction of 1.0 g of each of these supports in the solution containing PVP, methanol, Ruthenium (III) chloride hydrate, and water, followed by reduction with NaBH_4_, filtration, and drying at 90 °C overnight. All samples were consecutively treated at elevated temperatures in a commercial flow-bed microreactor (Linkam, Redhill, UK, CCR1000) at 450 °C (ramp 10 °C.min^−1^) for 2 h in a flow of 5 Ml min^−1^ consisting of 30 vol% H_2_ in He. Before opening and collecting the samples in vials, the flow composition was changed to 5 mL He min^−1^ and the samples were cooled to room temperature.

### 4.2. Characterization

A selection of samples after reductive thermal treatment was analyzed by transmission electron microscopy (TEM, Philips CM300ST-FEG, Amsterdam, The Netherlands) to determine the size, distribution, and morphology. Ex situ X-ray photoelectron spectroscopy (XPS, Quantera SXM, Physical Electronics, Chanhassen, MN, USA) using a monochromatic Al K_α_ x-ray source (1486.6 eV) was performed to determine the oxidation state of the Ru. The measurements were performed at 120 eV pass energy and 0.4 eV step size, and the working pressure in the chamber was typically lower than 7 × 10^−9^ Torr. The results were analyzed using the Multipak software (version 9.8) and the binding energy of all the spectra was calibrated using the adventitious carbon peak at 284.8 eV. UV-vis spectra were recorded using an Avantes probe and spectrometer (AvaSpec-2048, Apeldoorn, The Netherlands) to evaluate the absorbance spectra of the various catalysts. The metal loading was quantified by X-Ray Fluorescence spectroscopy (XRF, Philips PW 1480, Amsterdam, The Netherlands).

#### 4.2.1. Diffuse Reflectance Fourier Transform Infrared (DRIFT) Spectroscopy

DRIFT analysis was performed using a VERTEX 70 spectrometer (Bruker, Singapore) in combination with a diffuse reflectance spectroscopy cell (Harrick, Pleasantville, NY, USA, Praying Mantis, Bengaluru, India), equipped with a three-window dome and a temperature controller (Harrick, Anadis Instruments, Almere, The Netherlands). The measurements were performed at a spectral resolution of 4 cm^−1^. Before each measurement, the powdered samples of ~10 mg were pretreated in a flow of 50 mL min^−1^ containing 30 vol-% H_2_ in He, at 450 °C for 2 h, identical to the procedure followed prior to the activity measurements and sample characterization. This was carried out to ensure that any potentially present RuO_x_ was reduced to Ru and that any carbon was removed from the sample. During the cooling down of the sample, background spectra were recorded at 450 °C, 350 °C, 250 °C, and 50 °C. After cooling down to 50 °C, the reactor was purged with 50 mL min^−1^ He for 30 min to remove any remaining H_2_ gas. Then, a flow of 35 mL min^−1^ of 14.3% CO_2_ in He was introduced into the reactor for a duration of 30 min, to complete the interaction of the catalyst surface with CO_2_, which was confirmed by the recording of several spectra. Then, the first DRIFT spectrum of a series at each temperature was recorded. Subsequently, hydrogen was introduced (adding 15 mL min^−1^, total flow now 50 mL min^−1^ containing 10% CO_2_ and 30% H_2_), followed by 30 min of stabilization—and the second DRIFT spectrum was recorded. The gas flow was then closed, and light was subsequently introduced. A 120 W Hg lamp (HP-120, Dr. Grobel UV-Elektronik GmbH, Ettlingen, Germany, with the UV/Vis spectrum provided in Appendix A) was used to illuminate the samples via a fiber optic cable at a distance of ~6 cm from the quartz window of the three-window dome, resulting in exposure of the window of the dome to approximately 360 mW/cm^2^. A third DRIFT spectrum was recorded—after 30 min of illumination. Then, the gas flow was opened again at 35 mL min^−1^ containing 14.3% CO_2_ in He, and the temperature increased to 250 °C, followed by stabilization, recording of the first DRIFT spectrum at this temperature, introduction of 15 mL min^−1^ of H_2_, recording of the second spectrum, closure of the flow, illumination for 30 min, and recording of the third spectrum. This cycle was repeated with the same catalyst at 350 °C and 450 °C.

#### 4.2.2. (DRIFT) Spectroscopy During LED Illumination at 530 nm and 365 nm

The pretreatment of a fresh sample and introduction of gases of at each wavelength was identical to the UV/Vis experiments. Narrow band exposure to 365 nm (UV) or (maximum at) 530 nm (green) was independently investigated. The intensity vs. distance curves are shown in Appendix A—resulting in exposure of the window of the dome to ~7 mW/cm^2^ when positioning the UV-LED at ~1 cm away from the window, and ~8.5 mW/cm^2^ to green light, again positioned at ~1 cm away from the window.

In addition to a temperature-dependent series recording spectra during illumination in the presence of CO_2_ and H_2_, a complete temperature series was also recorded during illumination and exposure of the catalyst to CO_2_; thus without the addition and involvement of H_2_. After the introduction of CO_2_, the flow was discontinued, the first spectrum recorded in the dark, and LED illumination initiated for 30 min, after which a second spectrum was recorded during illumination. Then, the flow of gas (35 mL min^−1^, 14.6% CO_2_ in He) was reinitiated and the sample heated to the next desired temperature.

### 4.3. Photothermal Reactivity by Transient Analysis

Photothermal catalytic reduction of CO_2_ by H_2_ was investigated in a commercial flow-bed microreactor (Linkam, CCR1000, Redhill, UK) and CH_4_, CO, and H_2_O were detected by mass spectrometry (MS, PFEIFFER, OmniStar GSD320, Aßlar, Germany). The temperature was measured via an S-type platinum/rhodium thermocouple—positioned adjacent to the small ceramic cup—holding the sample. The thickness of the sample bed amounted to a few mm. First, 50 mg of sample was pretreated in situ in a 1.5 mL min^−1^ H_2_/3.5 mL min^−1^ He gas flow at 450 °C with a heating ramp of 10 °C min^−1^. Then, the catalyst was kept at this temperature for 2 h and subsequently cooled in the gas mixture to 50 °C. Again, this was done to reduce any potentially present RuO_x_ to Ru and to remove possible carbon contaminations from the sample. Subsequently, at this temperature, a reactant gas flow containing 0.5 mL min^−1^ CO_2_/1.5 mL min^−1^ H_2_/3 mL min^−1^ He was switched on. Purging with this gas flow was continued for 60 min. before heating up the samples. A 120 W Hg lamp (HP-120, Dr. Grobel UV-Elektronik GmbH, Ettlingen, Germany, with the spectrum provided in Appendix A) was used to illuminate the samples via a fiber optic cable at a distance of 6 cm, resulting in exposure of a quartz window of the microreactor from the top to an intensity of 360 mW/cm^2^.

To compare the intrinsic reactivities of different catalysts, kinetic measurements were first carried out between 120 and 220 °C, thus outside the range of equilibrium constraints. At each reaction temperature, the samples were kept for 30 min. in the dark, and then exposed to light-on, light off cycles for 30 min each.

The photothermal performance was studied further by increasing the temperature from 220 to 450 °C and then stepwise decreasing the temperature to 250 °C, to assess the presence of hysteresis and/or deactivation of the catalyst. At each temperature, the performance in the dark was analyzed for 20 min, followed by performance under irradiation for 20 min. This procedure was repeated three times. The photothermal stability of the Ru/TiO_2_ catalyst was also tested at 250 °C for 24 h by continuously alternating between dark and (UV-vis) irradiation conditions.

## 5. Conclusions

The experimental data provided in this study clearly support the following conclusions:The Ru/TiO_2_ catalyst used for methanation of CO_2_ shows a photo-response, which is larger than that of the catalysts reported in the literature. This is likely due to the strong metal (Ru) support (TiO_2_) interactions induced by the high temperature treatment in H_2_, preceding photothermal catalysis.Using a fixed illumination energy of 360 mW/cm^2^, the strongest light-induced enhancement in conversion of CO_2_ was obtained in the range of 200–250 °C. At 200 °C, the rate rapidly increases from ~1.2 mol g_Ru_^−1^ h^−1^ to ~1.8 mol g_Ru_^−1^ h^−1^ upon illumination. Light does not induce a change in mechanism given the similar activation energy, while the increase in rate would agree with a global temperature rise of the catalyst of ~10 °C.A change in selectivity from CH_4_ towards CO was observed at 450 °C—enhanced by illumination.DRIFT spectroscopic analysis in static gas conditions shows a diminishing intensity and shift in the Ru–CO absorption frequency upon illumination in isothermal conditions—indicative of light-induced desorption of CO equivalent to a temperature rise of several 100 s of degrees.Using the CO IR spectrum as an in-situ probe, localized heating (by Ru visible light absorption) cannot explain the observed decreasing CO coverage. Rather, interfacial charge transfer processes should play a role, in agreement with recent spectroscopic observations [36].To explain the limited increase in CH_4_ formation rate by non-thermal lowering of the CO coverage, we hypothesize that the surface concentrations of other adsorbates, relevant in the reaction mechanism, such as Ru–H, are also affected by illumination.

## Figures and Tables

**Figure 1 molecules-30-02577-f001:**
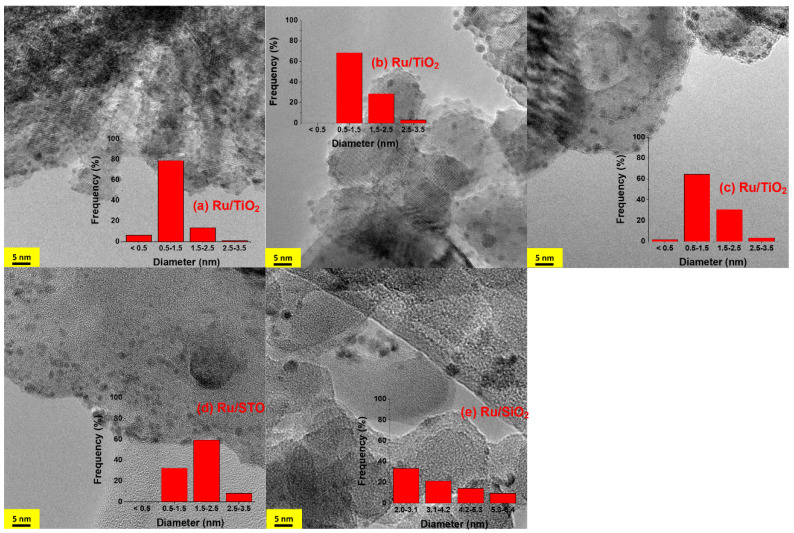
TEM images of Ru/TiO_2_ after: (**a**) H_2_ pretreatment; (**b**) reaction at 250 °C with on-off light cycles (20 min each) for a duration of 24 h; (**c**) stepwise increase in temperature from 250 to 450 °C and cool down to 250 °C; (**d**) Ru/STO; and (**e**) Ru/SiO_2_ after H_2_ pretreatment.

**Figure 2 molecules-30-02577-f002:**
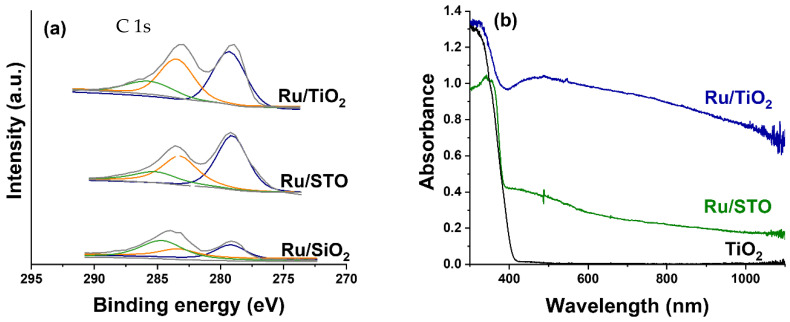
(**a**) XPS analysis of as-prepared Ru particles supported on TiO_2_, STO, and SiO_2_. The C 1s, Ru 3d 3/2, and Ru 3d 5/2 deconvolution corresponds to the green, yellow, and blue traces, respectively. (**b**) UV-vis absorbance spectra of the Ru/TiO_2_, Ru/STO, and TiO_2_ catalysts after H_2_ pretreatment at 450 °C.

**Figure 3 molecules-30-02577-f003:**
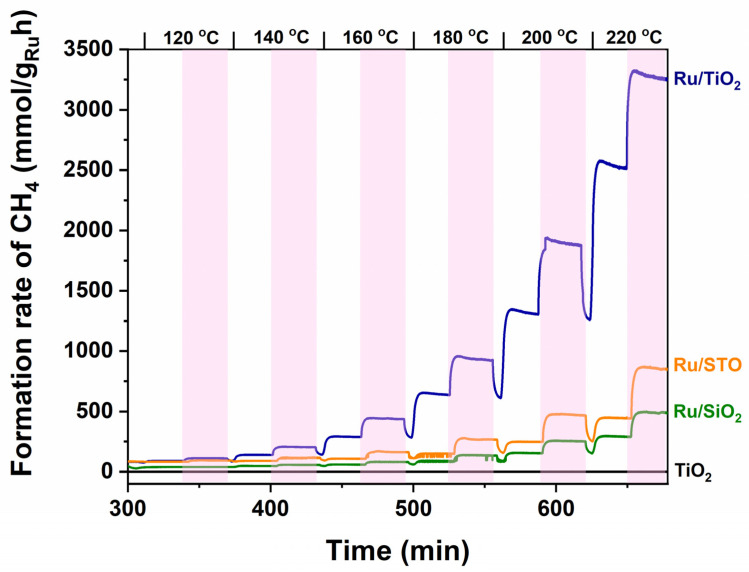
The Ru-normalized CH_4_ formation rates (left axis) from 120 °C to 220 °C (temperature indicated above graph) as determined for Ru/TiO_2_, Ru/STO, and Ru/SiO_2_ catalysts, respectively. Exposure to UV-vis irradiation is indicated by the pink rectangular areas—see the legend, as indicated in the Figure, for the respective catalysts. The TiO_2_ support does not show any activity—even when illuminated. Ilumination was provided according to the spectrum shown in Appendix A, with an intensity of 360 mW/cm^2^.

**Figure 4 molecules-30-02577-f004:**
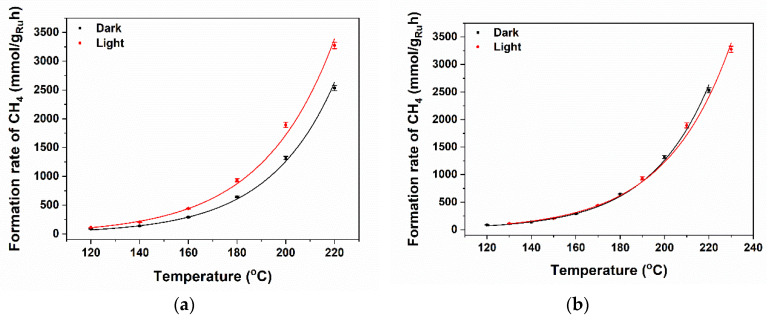
(**a**) The Ru-normalized CH_4_ formation rates (left axis) plotted as a function of temperature with light off (black) and light on (red curve). (**b**) The curves can be overlapped if the light-on curve is shifted to higher temperature values by ~10 °C. By comparing the Arrhenius plots (See Appendix A), the activation energies can be estimated to be quite similar in dark conditions and upon illumination—suggesting that light enhances the number of available sites. Based on the DRIFT analysis to follow, this is proposed to be due to changes in CO coverage.

**Figure 5 molecules-30-02577-f005:**
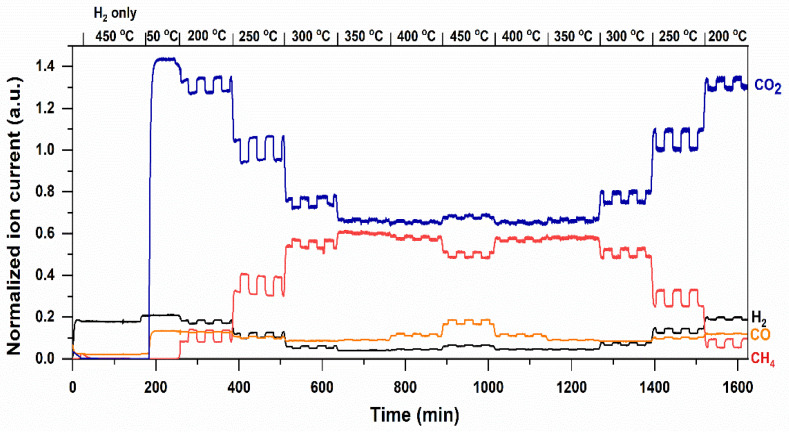
Mass spectrometry results (normalized ion current—*y*-axis) of photothermal hydrogenation of CO_2_ on Ru/TiO_2_ in the dark and under UV-vis irradiation, by stepwise increasing temperature from 50 to 450 °C, and decreasing from 450 to 200 °C, respectively. Temperature steps are indicated above the figure.

**Figure 6 molecules-30-02577-f006:**
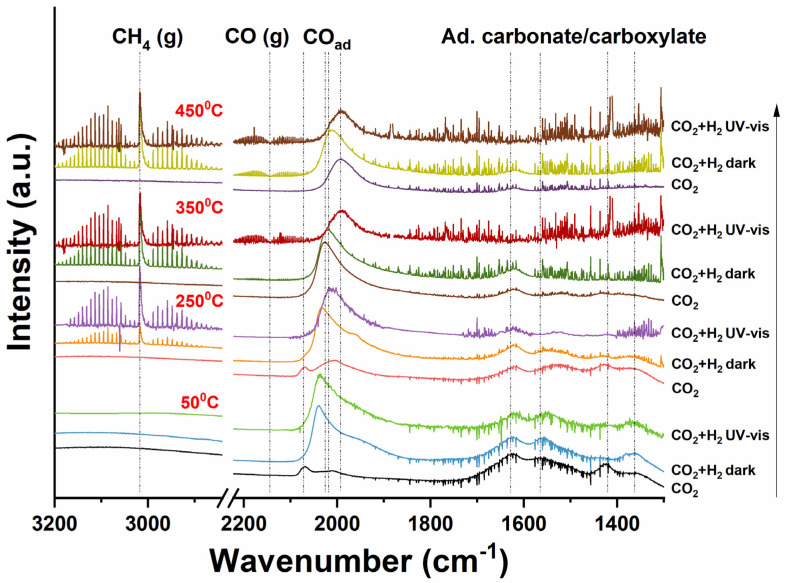
In-situ DRIFT spectra of photothermal CO_2_ reduction on Ru/TiO_2_ in the dark and under UV-vis irradiation. At each temperature (as indicated in the figure), three conditions are compared: exposure to CO_2_; exposure to H_2_/CO_2_; and exposure to H_2_/CO_2_ and UV-vis light.

**Figure 7 molecules-30-02577-f007:**
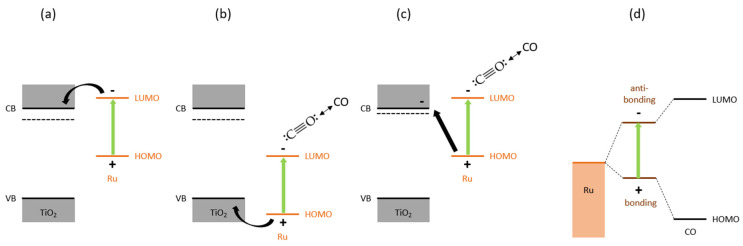
Charge transfer processes upon light activation of the Ru nanoparticles on TiO_x_. (**a**) and (**b**) illustrate photoinduced electron or hole transfer from the Ru towards the TiO_x_, depending on the position of the HOMO and LUMO energy levels of the Ru nanoclusters relative to the valence band (VB) and conduction band (CB) of the TiO_x_. (**c**) shows photoexcitation of the Ru nanoclusters, followed by the electron promoting CO desorption and hole transfer towards partly reduced TiO_x_ induced by the hydrogen treatment preceding the photothermal catalysis. (**d**) presents photoexcitation of a hybrid bonding state formed between Ru and CO towards the antibonding state, destabilizing the Ru–CO bond. The latter is unlikely to play a significant role here, as in that case Ru/SiO_2_ should have shown a higher performance.

**Figure 8 molecules-30-02577-f008:**
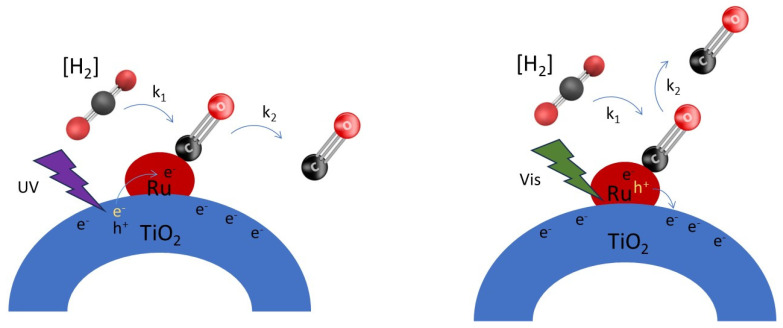
Effect of charge transfer processes on Ru–CO coverage—the **left** shows the situation upon UV light activation of the Ru nanoparticles on TiO_x_, and the **right** shows the situation upon green light activation of the Ru nanoparticles on TiO_x_.

**Table 1 molecules-30-02577-t001:** Comparison of performance data in this study and as reported in the literature for light activated Ru catalysts.

Catalyst	T (°C)	Reaction Rate (mmol g_Ru_^−1^ h^−1^)	Light Source	Light Intensity (mW/cm^2^)	Reaction Condition	Reference
Ru/TiO_2_	160	380 (CH_4_)	Hg lamp	360	gas flow	This work
Ru/STO	160	80 (CH_4_)	Hg lamp	360	gas flow	This work
Ru/SiO_2_	160	41 (CH_4_)	Hg lamp	360	gas flow	This work
Ru/Al_2_O_3_ RuO/Al_2_O_3_	150	135 (CH_4_) 85 (CH_4_)	Solar simulator	100	Batch	[31]
Ru/TiO_2_	150	172 (CH_4_)	Solar simulator	100	gas flow	[5]
RuO_2_/STO	150	50 (CH_4_)	Xe lamp	100	gas flow	[29]

## Data Availability

The original contributions presented in this study are included in the article/Appendix A. Further inquiries can be directed to the corresponding author.

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
