# Peer review of "Understanding the Light-Driven Enhancement of CO2 Hydrogenation over Ru/TiO2 Catalysts"

_molecules, 2025, doi:10.3390/molecules30122577_

Round 1

Reviewer 1 Report

Comments and Suggestions for Authors

 The capture and utilization of CO2 is an urgent research area particularly with the accumulation of CO2 in the atmosphere at an increasing rate, which causing various climate and environmental consequences. In the manuscript, the authors design a Ru/TiO2 catalysts that promoting the CO2 hydrogenation to CH4 under light irradiation. Although the work contains a number of a characterization techniques and testing of samples, however the experimental date is incomplete, a large number of contrast experiments need to be added. So the authors should further strengthen the manuscript. The manuscript might be accepted for publication with reversion after addressing the following issues.

  • The author needs to compare the differences in CO2 hydrogenation performance of metal Ru with different contents.
  • What form does Ru exist in? The current characterization cannot be determined, and the author needs to further test high-resolution electron microscopy.
  • Authors may carry out the ICP test to confirm the loading content of Ru on the prepared samples.
  • All the performance of CO2 hydrogenation in the manuscript should provide error bars. (Figure 4)
  • The authors should study the effect of different light intensities on its performance (under the same temperature). In addition, also need to study the effect of temperature on its performance. (under the same light intensity)
  • In this reaction, what light was functioning the photocatalysis, the author should discuss in detail.
  • To further determine whether the carbon source was derived from reactant. Therefore, an isotopic 13C measurement is crucial for the quantification of CH4. And the time course of isotopic labeling experiments for 13CO2 reduction should be provided (time profile of relative abundance of 13C labeled CH4).
  • Some latest research about photothermal CO2 reduction could be useful for discussion like Adv. Energy Mater. 2021, 11, 2002783; Nano Res., 2021, 14, 4828; Rare Met. 2022;41(5):1403–1405

Reviewer 2 Report

Comments and Suggestions for Authors

The manuscript entitled "Explaining Light-Induced Enhancement of CO2 Hydrogenation Rates using Ru/TiO2 Catalysts" investigates the effect of UV/visible light illumination on the catalytic performance of Ru/TiO2 in the hydrogenation of CO2 to methane. The authors report a significant increase in methane formation rates under light exposure, which they attribute not only to a modest increase in temperature but also to light-induced modifications in the electronic structure of Ru partially covered by TiO2. In situ DRIFT spectroscopy and the analysis of Ru–CO interactions under illumination provide valuable insight into the proposed mechanism. In my opinion, the satisfactory results and reasonable discussions prove it could be published in this journal. However, the following revisions should be considered.

P5 line 195: The authors propose the presence of a TiO2 overlayer on the Ru NPs based on a TEM micrograph (Fig S3). The image is not very clear, and the observation can also be explained by artifacts in the image (e.g. Fresnel fringes along the particle edge). Additional characterization is required to confirm the presence of a TiO2 overlayer.

P10 line 320: “As stated previously, we hypothesize the increase in CO2 conversion and CH4 productivity is due to a reduction in Ru-CO surface coverage.” This was not stated previously. Please clarify this hypothesis.

P10 line 340: “In addition, for Ru/TiO2 (Fig. 5) bands of low intensity appear at 2072 and 2018 cm-1, which can be attributed to surface-adsorbed CO on partly oxidized Ru and metallic Ru, respectively” Should this observation refer to Fig. 6 instead of Fig. 5? Please clarify how CO is adsorbed to Ru (via oxygen, via carbon, or via both atoms and motivate why). Furthermore, the fact that CO adsorbs to partly oxidized Ru and metallic Ru seems to contradict the earlier statement that the Ru NPs are covered by a TiO2 overlayer.

P11 line 359: “Most importantly, the Ru-CO band seems to decrease in intensity and significantly shifts to lower wavenumbers as a result of decreasing coverage with CO.” Explain how a decreasing coverage with CO results in a shift to lower wavenumbers. Does this affect the bond strength?

P11 line 376: “… is very similar to observed in Figure 5 …” Again, should this observation refer to Fig. 6 instead of Fig. 5?

P12 line 410: The reaction equations need to be revised to be consistent. Reaction (3) produces C(H)ad, while reaction (4) consumes Cad Adsorbed O(H) is variably denoted as O(H)ad or *O(H)

P12 line 425: “Our IR measurements at 350oC under pure CO2 show the dynamic equilibrium in action: the spectrum shows a clear band due to adsorbed CO, which we attribute CO2 dissociation (likely concurrently oxidizing Ti(III) vacancies, at the Ru/TiOx interface, or forming an OH and the reversible CO+O(H) recombination, which creates a steady state CO concentration but no net reaction (since CO is not a (major) product of the reaction).” This sentence is unclear and needs to be simplified. Also, the author mentions “Ti(III) vacancies” which, in principle, refers to a missing Ti(III) cation in the TiOx lattice. I expect the authors instead mean oxygen vacancies.

P15 line 456: The authors refer to “HOMO and LUMO energy levels of the Ru nanoclusters” However, metallic nanoparticles do not possess discrete molecular orbitals such as the HOMO and LUMO as found in individual molecules. Instead, they have a continuous electronic band that facilitates the collective oscillations of free electrons responsible for localized surface plasmon resonance. In other words, plasmonic NPs cannot have a HOMO or LUMO and vice versa. However, LSPR can still create “hot electrons” in the electronic band of the plasmonic metal, that are then injected into the conduction band of the semiconducting TiO2. This is the most likely scenario. Transfer of holes to TiO2 (scenario 2) or recombination of holes in Ru and electrons in TiO2 (scenario 3) is very unlikely because of the depletion zone, characteristic for a Schottky barrier at a Metal-Semiconductor interface, that lowers the concentration of electrons in the TiO2 conduction band near the interface.

P17 line 603: The authors should clarify why they expect a higher electron density on Ru (despite hot electron injection into TiO2) and motivate why a higher electron density (as opposed to a lower electron density) enhances the desorption of CO2.

Author Response

PLease see attachment (separate response file from reviewers 1 and 2)

Reviewer 3 Report

Comments and Suggestions for Authors
  1. The introduction section's review of photothermal CO₂ reduction to CH₄ and CO is overly simplistic. It should incorporate key findings from this study while discussing relevant literature to provide readers with more insightful context.
  2. The particle size distribution in Fig.1 may require re-evaluation with quantified size ranges. The amorphous overlayer observed in Fig.S3 needs clarification regarding whether it represents Ru nanoparticles or TiOx species. EDX mapping could provide supporting evidence.
  3. Comparative analysis of bandgapamong different materials should be included.  
  4. The choice of Hg lamp (with high UV content and intense light intensity) for performance comparison may not effectively demonstrate the material's advantages. Since Fig.S6 indicates minimal UV contribution to CH₄ production, using a solar-simulating Xe lamp would better showcase the material's merits.  
  5. Fig.6 should clearly label different surface species to facilitate reader comparison.  
  6. The attribution of activity differences solely to SMSI effects lacks rigorous evidence. The performance variation between Ru/STO and Ru/SiO₂ could alternatively stem from support oxygen vacancy concentration or metal-support interaction strength, yet no XPS valence state or oxygen defect analysis is provided.  
  7. Formatting issues require attention: some references lack page numbers and volume information (should follow journal style), superscript/subscript errors exist in the abstract, and unit notation inconsistencies (e.g., mL·min⁻¹, mol·gRu⁻¹·h⁻¹, mmol·gRu⁻¹·h⁻¹) need standardization.

Author Response

Please see response file (including the response to the second reviewer

Round 2

Reviewer 3 Report

Comments and Suggestions for Authors

The authors have revised this manuscript based on my comments. Thus I recommend it to be accepted by Molecules.

Author Response

We would like to thank the reviewer for the constructive suggestions, which have greatly improved the quality of this manuscript.